# The Effects of Topiroxostat, a Selective Xanthine Oxidoreductase Inhibitor, on Arterial Stiffness in Hyperuricemic Patients with Liver Dysfunction: A Sub-Analysis of the BEYOND-UA Study

**DOI:** 10.3390/biomedicines11030674

**Published:** 2023-02-23

**Authors:** Yuya Fujishima, Hitoshi Nishizawa, Yusuke Kawachi, Takashi Nakamura, Seigo Akari, Yoshiyuki Ono, Shiro Fukuda, Shunbun Kita, Norikazu Maeda, Satoshi Hoshide, Iichiro Shimomura, Kazuomi Kario

**Affiliations:** 1Department of Metabolic Medicine, Graduate School of Medicine, Osaka University, 2-2, Yamada-oka, Suita 565-0871, Osaka, Japan; 2Medical Affairs Department, Sanwa Kagaku Kenkyusho Co., Ltd., Nagoya 461-8631, Aichi, Japan; 3Department of Adipose Management, Graduate School of Medicine, Osaka University, 2-2, Yamada-oka, Suita 565-0871, Osaka, Japan; 4Department of Metabolism and Atherosclerosis, Graduate School of Medicine, Osaka University, 2-2, Yamada-oka, Suita 565-0871, Osaka, Japan; 5Division of Cardiovascular Medicine, School of Medicine, Jichi Medical University, 3311-1, Yakushiji, Shimotsuke 329-0498, Tochigi, Japan

**Keywords:** hyperuricemia, hypertension, arterial stiffness, xanthine oxidoreductase inhibitors, liver dysfunction

## Abstract

Background: The effects of uric acid (UA)-lowering therapy with xanthine oxidoreductase (XOR) inhibitors on the development of cardiovascular diseases remain controversial. Based on recent findings that plasma XOR activity increased in liver disease conditions, we conducted a sub-analysis of the BEYOND-UA study to examine the differential effects of topiroxostat on arterial stiffness based on liver function in hyperuricemic individuals with hypertension. Methods: Sixty-three subjects treated with topiroxostat were grouped according to baseline alanine aminotransferase (ALT) levels (above or below cut-off values of 22, 30, or 40 U/L). The primary endpoint was changes in the cardio-ankle vascular index (CAVI) from baseline to 24 weeks. Results: Significant reductions in CAVI during topiroxostat therapy occurred in subjects with baseline ALT ≥30 U/L or ≥40 U/L, and significant between-group differences were detected. Brachial-ankle pulse wave velocity significantly decreased in the ALT-high groups at all cut-off values. Reductions in morning home blood pressure and serum UA were similar regardless of the baseline ALT level. For eleven subjects with available data, ALT-high groups showed high plasma XOR activity, which was significantly suppressed by topiroxostat. Conclusions: Topiroxostat improved arterial stiffness parameters in hyperuricemic patients with liver dysfunction, which might be related to its inhibitory effect on plasma XOR.

## 1. Introduction

Hyperuricemia, defined as a serum uric acid (UA) level of more than 7.0 mg/dL, is a potential risk factor for life-threatening complications, such as chronic kidney disease (CKD) and cardiovascular disease (CVD) [1,2,3,4]. Individuals with hyperuricemia are a heterogeneous population, including reduced renal/extrarenal excretion type, over-production type, and mixed type [5]. In particular, the overproduction type in liver is associated with visceral fat-based metabolic syndrome, including type 2 diabetes mellitus, hypertension, dyslipidemia, and nonalcoholic fatty liver disease/steatohepatitis (NAFLD/NASH) [6], all of which are major independent risk factors for CVD.

Xanthine oxidoreductase (XOR) is a rate-limiting enzyme that catalyzes the production of UA from hypoxanthine and xanthine, and is the pharmacological target of anti-hyperuricemic agents such as allopurinol, febuxostat, and topiroxostat. XOR generates reactive oxygen species (ROS) through its catabolic processes and can bind to the apical surface of vascular endothelial cells [7,8], suggesting a possible role of this enzyme in endothelial dysfunction [9,10]. Human XOR expression is detected mainly in liver, lungs, and gut [11]. Recently, we reported that circulating XOR in humans and mice markedly increased with elevations in liver enzymes such as serum alanine aminotransferase (ALT) or aspartate aminotransferase (AST), reflecting excessive leakage of hepatic XOR, and that topiroxostat, a selective XOR inhibitor, suppressed plasma XOR activity and attenuated the development of vascular neointima formation in a diet-induced mouse model of NAFLD/NASH [12,13]. Therefore, we hypothesize that XOR inhibitors may have the potential to prevent or delay cardiovascular complications, especially in patients with liver dysfunction, possibly beyond their UA-lowering effect.

Arterial stiffness is generally recognized as a predictor of cardiovascular morbidity and mortality, and increased arterial stiffness reflects structural and functional changes in the diffuse medial layer of the vessel wall [14,15]. The Beneficial Effect by Xanthine Oxidase Inhibitor on Endothelial Function Beyond Uric Acid (BEYOND-UA) study was the first randomized, controlled trial comparing the effects of the XOR inhibitors, topiroxostat and febuxostat, on arterial stiffness parameters in Japanese hypertensive patients with hyperuricemia [16]. In the overall analysis, there were no significant changes in the cardio-ankle vascular index (CAVI) or brachial-ankle pulse wave velocity (baPWV) during the 24-week follow-up period in either treatment group. However, based on our previous work [12], subjects with liver dysfunction (who are expected to have increased circulating XOR activity) may benefit most from treatment with XOR inhibitors. Of the currently available XOR inhibitors, topiroxostat was reported to have the strongest inhibitory effect against human plasma XOR activity, with a 50% inhibitory concentration (IC50) that was 194-fold and 16-fold lower than that of oxypurinol and febuxostat, respectively [17]. In the BEYOND-UA study, XOR activity significantly decreased during the treatment with topiroxostat, but not with febuxostat [16]. This post hoc subgroup analysis of the BEYOND-UA study determined the differential effects of topiroxostat on arterial stiffness in subgroups of hyperuricemic patients with hypertension based on baseline liver function.

## 2. Materials and Methods

### 2.1. Study Design

The BEYOND-UA study was a multicenter (n = 31), randomized, comparative, open-label, parallel trial conducted in Japan between March 2018 and December 2019 [16]. As previously described [15], the study enrolled patients aged 30–80 years who had hyperuricemia (serum UA ≥7 mg/dL; untreated or treated with allopurinol), hypertension that had been treated with stable antihypertensive therapy for ≥3 months, and a CAVI of ≥8 and ≤12. Exclusion criteria were as follows: History of hypersensitivity to trial drugs or allopurinol; treatment with anti-hyperuricemic drugs during the study or within 4 weeks prior to enrollment; existing cancer diagnosis; gouty arthritis within 2 weeks before enrollment;AST or ALT >2 times the upper limit of normal; serious liver dysfunction (Child-Pugh class B or C); renal dysfunction (estimated glomerular filtration rate <30 mL/min/1.73 m^2^); severe heart failure (New York Heart Association class 3 or 4); history of acute coronary syndrome or stroke within the previous 3 months; and participation in another clinical trial within the previous 6 months.

### 2.2. Randomization and Intervention

Patients were randomly assigned to receive treatment with topiroxostat or febuxostat for 24 weeks. Aiming to maintain serum UA at <6 mg/dL, topiroxostat was started at 40 mg/day then titrated to 80 mg/day at week 4, to 120 mg/day at week 8, and up to a maximum of 160 mg/day during weeks 8–24, and febuxostat was started at 10 mg/day titrated to 20 mg/day at week 4, 40 mg/day week 8 and up to a maximum of 60 mg/day during weeks 8–24.

As previously reported [16], 67 patients in the topiroxostat group and 68 in the febuxostat group were eligible for final safety and efficacy analysis. In the current sub-analysis, patients treated with topiroxostat were divided into two groups according to their baseline ALT level: above or below 22 U/L (the median value) (Table 1), above or below 30 U/L (Table 2), or above or below 40 U/L (the upper limit of normal) (Table 3) (Appendix A).

### 2.3. Outcomes

The primary endpoint was the change in CAVI from baseline to 24 weeks. Secondary outcomes were as follows: change in CAVI from baseline to 12 weeks; change in baPWV from baseline to 12 and 24 weeks; change in serum UA from baseline to 4, 8, 12, and 24 weeks; change in home blood pressure (BP) from baseline to 4, 8, 12, and 24 weeks; change in the urinary albumin-creatinine ratio (UACR) from baseline to 12 and 24 weeks. Change in plasma XOR activity from baseline to 12 and 24 weeks was investigated as an exploratory endpoint (n = 11 for the topiroxostat group).

### 2.4. Assessments

CAVI was measured at baseline, and after 12 and 24 weeks of treatment using a CAVI device (Vasera VS3000). Examinations were performed after a 5-min rest period. The pressure of all cuffs was kept at 50 mmHg to minimize the effect of cuff pressure on hemodynamics. BP was then measured. CAVI was determined using the following formula: CAVI = a [(2ρ/ΔP) × ln (Ps/Pd) PWV2] + b, where a and b are constants, ρ is blood density, ΔP is Ps − Pd, Ps is systolic BP, Pd is diastolic BP, and PWV is pulse wave velocity.

PWV was determined by dividing the vascular length by the time (T) taken for the pulse wave to travel from the aortic valve to the ankle. However, in practice, T was difficult to obtain because the time the blood left the aortic valve was difficult to identify from the sound of the valve opening. Therefore, because the time between the sound of the aortic valve closing and the notch of the brachial pulse wave is theoretically equal to the time between the sound of the aortic valve opening and the rise of the brachial pulse wave, T was determined by adding the time between the sound of the aortic valve closing and the notch of the brachial pulse wave, and the time between the rise of the brachial pulse wave and the rise of the ankle pulse wave.

Home BP was measured at baseline and after 4, 8, 12, and 24 weeks of treatment using a cuff oscillometric device (HEM-7080-IC; Omron Healthcare Co., Ltd., Kyoto, Japan). All measurements were performed according to the latest guidelines available at the time the trial was conducted [18]. Patients were instructed to measure their morning home BP (two readings within 1 h after waking, taken after urination, before taking morning medications and after 1–2 min of seated rest) on five successive days immediately prior to their scheduled clinic visit.

Plasma XOR activity measurement was performed by Sanwa Kagaku Kenkyusho Co., Ltd. (Inabe, Japan), using liquid chromatography/triple quadrupole mass spectrometry (LC/TQMS; Nexera HLC (SHIMADZU, Kyoto, Japan)/QTRAP 4500 (SCIEX, MA, USA)) to detect [13C2,15N2]-uric acid using [13C2,15N2]-xanthine as a substrate, as previously described [19].

### 2.5. Statistical Analysis

Mixed-effects model repeated measures (MMRM) analysis was used to compare the changes in CAVI and other outcomes from baseline to week 4, week 8, week 12, and week 24. MMRM included the subgroup based on each ALT cut-off value, time points (0, 4, 8, 12, and 24 weeks), the interaction between the group and time points as fixed effects, and age and sex as covariates. A two-sided test was used, and *p*-values of <0.05 were considered statistically significant. Intergroup comparisons were tested with a t-test for continuous variables, and Pearson’s Chi-squared test or Fisher’s exact test was used for dichotomous data. Data were analyzed using SAS version 9.4 (SAS Institute) at the Jet Academy, Tokyo, Japan.

## 3. Results

### 3.1. Baseline Characteristics of Study Subjects Treated with Topiroxostat

Of the 67 subjects treated with topiroxostat, baseline CAVI data were not obtained in 4 patients; the remaining 63 were included in this post hoc analysis (Appendix A). Two patients were switched from previous allopurinol therapy. Table 1, Table 2 and Table 3 show baseline clinical characteristics in patient subgroups based on each ALT cut-off. Both ALT and AST levels were significantly higher in each ALT-high subgroup, and there was a tendency for body mass index (BMI) to be higher (*p* = 0.074) and the use of calcium channel blockers (CCBs) more frequent (*p* = 0.066) in the ALT ≥40 versus <40 U/L group. No significant differences between the ALT-high and -low groups at each cut-off value were seen for other clinical parameters, including serum UA, systolic and diastolic BP, CAVI, and baPWV (Table 1, Table 2 and Table 3).

### 3.2. Arterial Stiffness

Figure 1 shows the time course changes in arterial stiffness during the study period, assessed by CAVI or baPWV. As shown in Figure 1B,C, significant reductions in CAVI were observed at week 24 in patients with baseline ALT >30 U/L or >40 U/L. On the other hand, in the ALT-low groups, CAVI increased slightly but significantly in patients with baseline ALT <22 U/L (Figure 1A) and remained unchanged in the ALT <30 U/L or ALT <40 U/L groups (Figure 1B,C). At week 24, baPWV was significantly decreased in the ALT-high groups at all three cut-offs, whereas there was no significant change in each of the ALT-low groups (Figure 1D–F). There were also significant between-group differences in changes from baseline to week 24 in both CAVI (Figure 1A–C) and baPWV (Figure 1D–F).

### 3.3. Morning Home Blood Pressure

Overall, morning home systolic (Figure 2A–C) and diastolic (Figure 2D–F) BP decreased significantly from baseline, irrespective of baseline ALT level. There were no between-group differences in BP changes, except for at week 24 when patients with ALT ≥40 U/L had greater reductions in systolic and diastolic BP compared to those with ALT <40 U/L (Figure 2C,F).

### 3.4. Uric Acid Levels and Plasma XOR Activity

Serum UA levels decreased significantly from baseline to week 24 in all ALT subgroups (*p* < 0.001) (Figure 3A–C). However, there was a trend toward smaller reductions in serum UA in patients with ALT ≥30 U/L versus <30 U/L (Figure 3B). For the 11 subjects with available data, baseline plasma XOR activity increased gradually in the subgroups with higher baseline ALT as the cut-off value increased from 22 to 40 U/L (Figure 4A–C). Elevated XOR activity in the ALT-high groups decreased to levels that were similar to those in the ALT-low groups at week 12 and week 24 of treatment with topiroxostat (Figure 4A–C).

### 3.5. Urinary Albumin-Creatinine Ratio

In the overall population, UACR decreased significantly from baseline during topiroxostat treatment (16), but there were no significant changes in UACR in all the subgroups nor between-group differences (Figure 5A–C).

### 3.6. Alanine Aminotransferase

The time course changes in serum ALT levels are shown in Figure 6A–C. There were no significant changes in ALT nor between-group differences at cut-offs of ALT 22 U/L and 30 U/L, whereas patients with baseline ALT levels higher than 40 U/L showed a significant decrease in ALT at week 4 (*p* < 0.05) and week 24 (*p* < 0.0001) (Figure 6C).

## 4. Discussion

To the best of our knowledge, this sub-analysis of the BEYOND-UA trial is the first study to show that the selective XOR inhibitor topiroxostat improved arterial stiffness parameters (CAVI and baPWV) in hyperuricemic patients with hypertension and liver dysfunction.

Evidence from epidemiological studies suggests that elevated serum UA levels are a risk factor for CKD and CVD (1–4). Generation of XOR-dependent vascular ROS has been considered one of the underlying mechanisms responsible for endothelial dysfunction and atherosclerosis associated with hyperuricemia [9,10]. However, it remains controversial whether UA-lowering therapy with XOR inhibitors is effective for preventing CVD development in individuals with hyperuricemia. In two randomized trials, treatment with allopurinol attenuated the progression of carotid intima-media thickness (IMT) in patients with asymptomatic hyperuricemia and type 2 diabetes [20] or recent ischemic stroke [21]. In the Febuxostat for cerebral and caRdiovascular Events prevention study (FREED), in which 1070 patients with hyperuricemia at high CVD risk were randomly allocated to treatment with febuxostat or conventional therapy, the primary composite event rate (cerebral, cardiovascular, and renal events, and all deaths) was significantly lower in the febuxostat versus control group, but this result was mainly driven by differences in renal impairment [22]. Furthermore, a recent prospective, randomized, open-label, blinded trial, enrolling 5721 subjects with ischemic heart disease showed no difference in the rate of the primary outcomes of non-fatal myocardial infarction, non-fatal stroke, or cardiovascular death between patients receiving allopurinol and those receiving usual care [23]. These mixed results highlight the importance of identifying potential factors that might affect the cardiovascular protective effect of XOR inhibitors.

Accumulating clinical data indicate that NAFLD, the most common liver disease worldwide, increases the risk of CVD, independent of established cardiovascular risk factors [24,25,26]. Moreover, certain studies have suggested that liver function tests themselves, including ALT, AST, gamma-glutamyltransferase (GGT), and alkaline phosphatase (ALP), can be potential CVD risk markers, independent of their relationship to NAFLD [27,28,29]. Our previous studies in both humans and mice demonstrated that increased plasma XOR activity was directly induced by liver damage, together with increases in liver enzymes such as serum ALT and AST. Moreover, high XOR in liver disease conditions accelerated purine catabolism in the plasma per se using hypoxanthine secreted from vascular endothelial cells or adipocytes as substrate, which was accompanied by the development of vascular endothelial injury and neointimal proliferation [11,12,13]. Additionally, a cross-sectional study by another group reported a significant positive correlation between plasma XOR activity and CAVI in patients with type 2 diabetes and liver dysfunction [30]. These results suggest pathological crosstalk from the damaged liver to vascular diseases via hepatic XOR. 

In that context, the aim of this post hoc analysis was to assess the efficacy of XOR inhibitors on arterial stiffness in subjects with liver dysfunction. In addition to the ALT cut off values of 22 U/L (the median value) and 40 U/L (the upper limit of normal), we also analyzed our subjects by dividing them above or below 30 U/L because such a slight increase of ALT ≥30 U/L in Japanese subjects has been reported to be associated with lifestyle-related chronic liver diseases such as NAFLD [31]. Baseline levels of CAVI and baPWV in the ALT-high groups tended to be higher as the ALT cut-off value increased (Figure 1). After 24 weeks’ treatment with topiroxostat, we found that CAVI significantly decreased from baseline in subjects with baseline ALT ≥30 U/L (mean ALT 47.8 U/L) and ≥40 U/L (mean ALT 61.1 U/L), and baPWV decreased significantly from baseline in subjects with ALT ≥22 U/L (mean ALT 35.8 U/L), ≥30 U/L, and ≥40 U/L. Albeit in a limited number of subjects, patients with a high baseline ALT level also showed increased plasma XOR activity, which was markedly suppressed during treatment with topiroxostat (Figure 4). Given that serum UA significantly decreased to approximately 6 mg/dL or less, irrespective of baseline ALT level (Figure 3), such improvement in arterial stiffness markers during topiroxostat therapy in the ALT-high groups was assumed to have resulted not only from a reduction in serum UA but also from a reduction in plasma XOR activity per se. In our subjects treated with febuxostat (n = 65), in whom no significant change was observed in plasma XOR activity [16], post hoc analysis using the same method did not detect any changes in either CAVI or baPWV (data not shown). On the other hand, in a recently published sub-analysis of the PRIZE (program of vascular evaluation under uric acid control by xanthine oxidase inhibitor, febuxostat: multicenter, randomized controlled) study, long-term (24 months’) treatment with febuxostat significantly improved arterial stiffness (assessed by CAVI or baPWV) compared with non-pharmacological management, without any change in carotid IMT progression [32]. Thus, further large-scale studies are needed to conclude whether differences in plasma XOR inhibitory activity between different XOR inhibitors influence their effects on arterial stiffness, especially in patients with liver dysfunction.

In the overall analysis of the BEYOND-UA study, there was a significant reduction in UACR over 24 weeks’ treatment with topiroxostat, especially in subjects with microalbuminuria at baseline [16]. However, unlike the CAVI and baPWV results, there were no differences in serial changes in UACR based on baseline ALT level (Figure 5). The underlying mechanisms for this discrepancy are uncertain, but similar reductions in systolic and diastolic BP in the ALT-high and ALT-low groups (Figure 2) might have a relatively strong beneficial effect on UACR. High UA is assumed to be involved in the development of hypertension and renal vasoconstriction via activation of the renin-angiotensin system (RAS) and reducing endothelial nitric oxide bioavailability [4,33]. As such, it is possible that improved hypertensive status associated with the reduction in UA had a greater impact on the change in UACR during treatment than the decrease in XOR activity. On the other hand, arterial stiffness is related not only to BP but also to remodeling of arterial structures caused by the proliferation of vascular smooth muscle cells (VSMCs) and connective tissues [14,15]. We previously showed that liver-derived XOR stimulated the proliferation and dedifferentiation of in vitro human VSMCs and in vivo neointima formation composed of proliferative SMCs in diet-induced NAFLD/NASH model mice, both of which were attenuated by the treatment with topiroxostat [13]. Because subjects in the ALT-high groups had markedly elevated baseline plasma XOR activity, the beneficial effects of topiroxostat on VSMCs via XOR suppression might result in a differential arterial stiffness response based on liver function.

Although not a predefined outcome in the BEYOND-UA study, changes in serum ALT were retrospectively examined to confirm the effect of topiroxostat on liver function itself. There were no significant changes in ALT in the ALT ≥22 U/L (Figure 6A) and ALT ≥30 U/L (Figure 6B) groups during the study period, suggesting that reductions in CAVI or baPWV and suppression of plasma XOR activity were not mediated by improved liver function, at least in these two subgroups. On the other hand, we found a significant decrease in ALT levels in subjects with baseline ALT ≥40 U/L (Figure 6C). Because of the small number of subjects (n = 8), the possibility that this is a chance finding cannot be ruled out. However, previous experimental and clinical studies provide evidence that XOR inhibitors have the potential to prevent the development of NAFLD, through attenuation of hepatic lipid accumulation, insulin resistance, and activation of macrophage and NLRP3 inflammasome [34,35,36]. Thus, these beneficial effects of topiroxostat on liver function might contribute, in part, to the improvement in arterial stiffness parameters in the ALT ≥40 U/L group.

There are several limitations in the present study. First, it is post hoc and exploratory by nature, and statistical comparisons were not adjusted for multiplicity. Although the data used in this study were generated in a randomized controlled clinical trial, the post hoc analysis was not pre-specified. Thus, potential sources of bias cannot be ruled out due to the post hoc nature of this analysis. Second, the results must be interpreted with caution due to the small sample size in each patient subgroup and due to the lack of a placebo control group. Third, the BEYOND-UA study was not originally designed to focus on liver function; therefore, the underlying causes of liver injury in each subject were not examined in detail, and subjects with different types of liver disease, not only NAFLD, could be enrolled in this study. In fact, we have confirmed that there was one subject with chronic hepatitis C whose ALT level was 15 U/L and two subjects with alcoholic hepatitis whose ALT levels were 22 and 28 U/L in the topiroxostat group. Moreover, we could not evaluate the outcomes in patients with severe liver dysfunction because an ALT or AST >2 times the upper limit of normal was an exclusion criterion for the main BEYOND-UA study. However, if inhibition of plasma XOR does mediate reductions in CAVI and baPWV, there is also a possibility that such patients might experience even more pronounced improvements in these arterial stiffness parameters during treatment with topiroxostat, but this remains to be determined.

## 5. Conclusions

Topiroxostat decreased CAVI and baPWV in hyperuricemic subjects who had higher ALT values at baseline, and this was accompanied by significant suppression of increased plasma XOR activity. Taken together, our results raise the possibility that topiroxostat has therapeutic potential for improving arterial stiffness and preventing atherosclerotic diseases in patients with liver dysfunction. Future large-scale prospective placebo-controlled studies enrolling NAFLD patients are warranted to confirm this hypothesis.

## Figures and Tables

**Figure 1 biomedicines-11-00674-f001:**
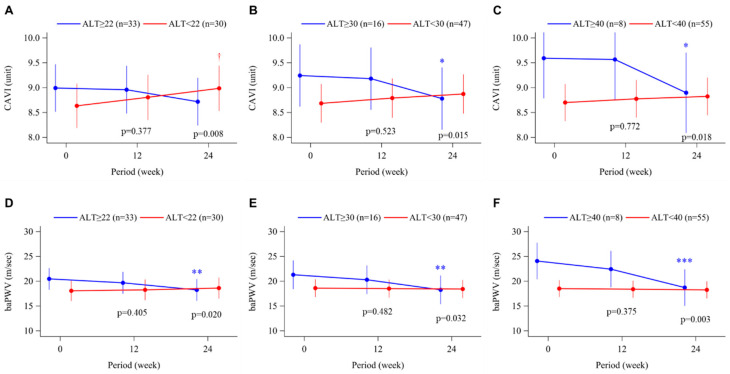
Time course of changes in the cardio-ankle vascular index (CAVI; **A**–**C**) and brachial-ankle pulse wave velocity (baPWV; **D**–**F**) from baseline to week 24 in the topiroxostat-treated group. Patients were divided into groups according to baseline alanine aminotransferase (ALT) level, above or below 22 U/L (median value; **A**,**D**), 30 U/L (**B**,**E**), and 40 U/L (upper limit of normal; **C**,**F**). Points and bars represent the least-squares mean and 95% confidence interval values using a mixed-effects model with repeated measures adjusted for sex and age. *p*-values within each graph are for the between-group difference in change from baseline. * *p* < 0.05, ** *p* < 0.01 and *** *p* < 0.001 for change from baseline in the ALT-high group. † *p* < 0.05 for change from baseline in the ALT-low group.

**Figure 2 biomedicines-11-00674-f002:**
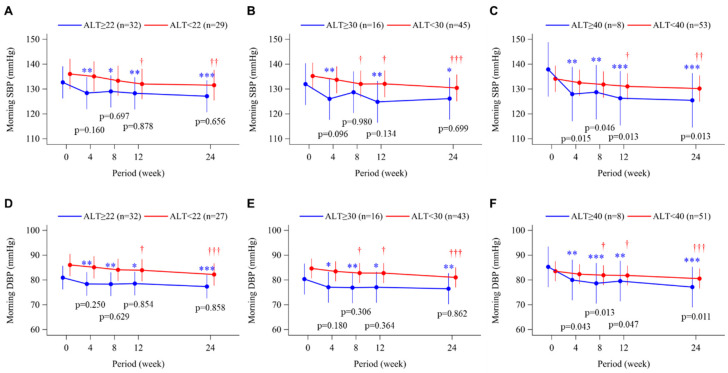
Time course of changes in morning home systolic blood pressure (SBP; **A**–**C**) and diastolic blood pressure (DBP; **D**–**F**) from baseline to week 24 in the topiroxostat-treated group. Patients were divided into groups according to baseline alanine aminotransferase (ALT) level, above or below 22 U/L (median value; **A**,**D**), 30 U/L (**B**,**E**), and 40 U/L (upper limit of normal; **C**,**F**). Points and bars represent the least-squares mean and 95% confidence interval values using a mixed-effects model with repeated measures adjusted for sex and age. p-values within each graph are for the between-group difference in change from baseline. * *p* < 0.05, ** *p* < 0.01 and *** *p* < 0.001 for change from baseline in the ALT-high group. † *p* < 0.05, †† *p* < 0.01 and ††† *p* < 0.001 for change from baseline in the ALT-low group.

**Figure 3 biomedicines-11-00674-f003:**
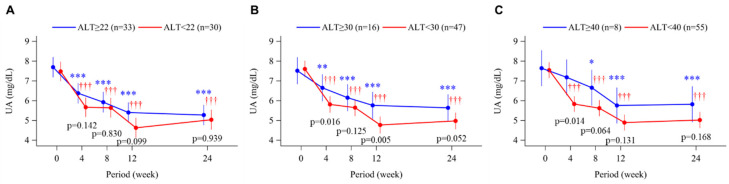
Time course of changes in serum uric acid from baseline to week 24 in the topiroxostat-treated group. Patients were divided into groups according to baseline alanine aminotransferase (ALT) level, above or below 22 U/L (median; **A**), 30 U/L (**B**), and 40 U/L (upper limit of normal; **C**). Points and bars represent the least-squares mean and 95% confidence interval values using a mixed-effects model with repeated measures adjusted for sex and age. p-values within each graph are for the between-group difference in change from baseline. * *p* < 0.05, ** *p* < 0.01, and *** *p* < 0.001 for change from baseline in the ALT-high group. ††† *p* < 0.001 for change from baseline in the ALT-low group.

**Figure 4 biomedicines-11-00674-f004:**
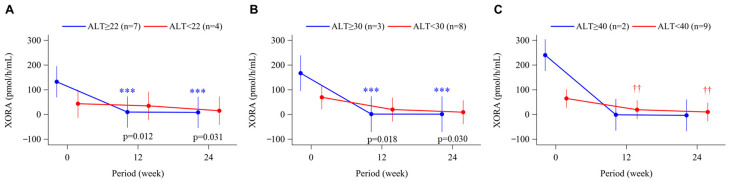
Time course of changes in plasma xanthine oxidoreductase activity (XORA) from baseline to week 24 in the topiroxostat-treated group. Patients were divided into groups according to baseline alanine aminotransferase (ALT) level, above or below 22 U/L (median; **A**), 30 U/L (**B**), and 40 U/L (upper limit of normal; **C**). Points and bars represent the least-squares mean and 95% confidence interval using the mixed-effects model with repeated measures adjusted for sex and age. p-values within each graph are for the between-group difference in change from baseline. *** *p* < 0.001 for change from baseline in the ALT-high group. †† *p* < 0.01 for change from baseline in the ALT-low group.

**Figure 5 biomedicines-11-00674-f005:**
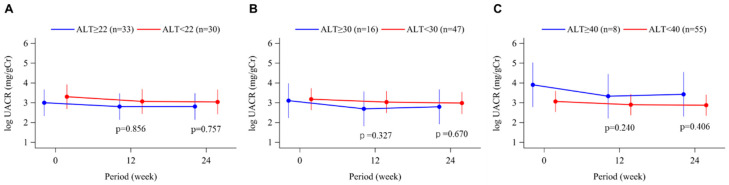
Time course of changes in the urinary albumin-creatinine ratio (UACR) from baseline to week 24 in the topiroxostat-treated group. Patients were divided into groups according to baseline ALT level, above or below 22 U/L (median; **A**), 30 U/L (**B**), and 40 U/L (upper limit of normal; **C**). Points and bars represent the least-squares mean and 95% confidence interval values using a mixed-effects model with repeated measures adjusted for sex and age. p values within each graph are for the between-group difference in change from baseline. UACR, urinary albumin-creatinine ratio.

**Figure 6 biomedicines-11-00674-f006:**
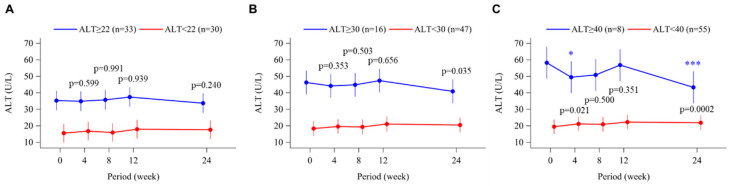
Time course of changes in alanine aminotransferase (ALT) from baseline to week 24 in the topiroxostat-treated group. Patients were divided into groups according to baseline ALT level, above or below 22 U/L (median; **A**), 30 U/L (**B**), and 40 U/L (upper limit of normal; **C**). Points and bars represent the least-squares mean and 95% confidence interval values using a mixed-effects model with repeated measures adjusted for sex and age. *p*-values within each graph are for the between-group difference in change from baseline. * *p* < 0.05 and *** *p* < 0.001 for change from baseline in the ALT-high group.

**Table 1 biomedicines-11-00674-t001:** Baseline clinical characteristics of participants with alanine aminotransferase level <22 U/L or ≥22 U/L.

Variables	ALT <22 U/L (n = 30)	ALT ≥22 U/L (n = 33)	*p*-Value
Age, years	68.9 ± 7.0	67.5 ± 7.8	0.482
Male, n (%)	23 (76.7)	30 (90.9)	0.172
Body mass index, kg/m^2^	25.5 ± 3.9	26.1 ± 2.9	0.453
Smoking, n (%)	6 (20.0)	7 (21.2)	1.000
Drinking, n (%)	18 (60.0)	26 (78.8)	0.169
Medical history, n (%)			
Diabetes mellitus	8 (26.7)	5 (15.2)	0.353
Dyslipidemia	11 (36.7)	13 (39.4)	1.000
Chronic kidney disease	0 (0.0)	2 (6.1)	0.493
Liver disease	1 (3.3)	2 (6.1)	1.000
Stroke	1 (3.3)	1 (3.0)	1.000
Heart failure	2 (6.7)	0 (0.0)	0.223
Coronary artery disease	1 (3.3)	4 (12.1)	0.357
Non-valvular atrial disease	1 (3.3)	3 (9.1)	0.614
Antihypertensives, n (%)			
ACEi	2 (6.7)	1 (3.0)	0.601
ARB	19 (63.3)	26 (78.8)	0.264
CCB	16 (53.3)	19 (57.6)	0.803
Beta-blocker	4 (13.3)	8 (24.2)	0.344
Diuretic	7 (23.3)	8 (24.2)	1.000
Other	1 (3.3)	1 (3.0)	1.000
Antidiabetic therapy, n (%)	8 (26.7)	5 (15.2)	0.353
Morning home SBP, mmHg	136.2 ± 15.5	133.1 ± 12.2	0.396
Morning home DBP, mmHg	86.0 ± 12.6	81.2 ± 11.4	0.129
Morning home HR, beats/min	69.9 ± 10.3	67.8 ± 11.0	0.463
CAVI, unit	8.9 ± 1.0	9.3 ± 1.6	0.236
baPWV, m/sec	18.7 ± 2.3	21.1 ± 8.6	0.132
ALT, U/L	15.3 ± 4.3	35.8 ± 17.6	<0.001
AST, U/L	21.0 ± 6.1	31.2 ± 11.6	<0.001
Uric acid, mg/dL	7.6 ± 1.0	7.8 ± 1.3	0.331
Creatinine, mg/dL	0.90 ± 0.25	0.92 ± 0.19	0.752
eGFR, mL/min/1.73 m^2^	64.9 ± 18.6	64.7 ± 14.0	0.967
hs-CRP, ng/mL	773 (428, 1500)	1100 (375, 1910)	0.995
NT-pro BNP, pg/mL	79 (45, 153)	51 (27, 139)	0.271
UACR, mg/g·Cr	20.6 (11.2, 42.4)	12.4 (7.9, 49.2)	0.405
Cystatin-C, mg/L	1.07 ± 0.21	1.04 ± 0.23	0.680

Values are mean ± standard deviation, median (interquartile range), or number of patients (%). Abbreviations: ACEi, angiotensin-converting enzyme inhibitor; ARB, angiotensin II receptor blocker; CCB, calcium channel blocker; SBP, systolic blood pressure; DBP, diastolic blood pressure; HR, heart rate; CAVI, cardio-ankle vascular index; baPWV, brachial-ankle pulse wave velocity; ALT, alanine aminotransferase; AST, aspartate aminotransferase; eGFR, estimated glomerular filtration rate; hs-CRP, high sensitive C-reactive protein; NT-proBNP, amino terminal-pro B-type natriuretic peptide; UACR, urinary albumin-creatinine ratio.

**Table 2 biomedicines-11-00674-t002:** Baseline clinical characteristics of participants with alanine aminotransferase level <30 U/L or ≥30 U/L.

Variables	ALT <30 U/L (n = 47)	ALT ≥30U/L (n = 16)	*p*-Value
Age, years	69.2 ± 6.5	65.2 ± 9.2	0.121
Male, n (%)	38 (80.9)	15 (93.8)	0.429
Body mass index, kg/m^2^	25.5 ± 3.3	26.7 ± 3.6	0.277
Smoking, n (%)	10 (21.3)	3 (18.8)	1.000
Drinking, n (%)	33 (70.2)	11 (68.8)	1.000
Medical history, n (%)			
Diabetes mellitus	10 (21.3)	3 (18.8)	1.000
Dyslipidemia	18 (38.3)	6 (37.5)	1.000
Chronic kidney disease	2 (4.3)	0 (0.0)	1.000
Liver disease	3 (6.4)	0 (0.0)	0.564
Stroke	1 (2.1)	1 (6.3)	0.446
Heart failure	2 (4.3)	0 (0.0)	1.000
Coronary artery disease	3 (6.4)	2 (12.5)	0.594
Non-valvular atrial disease	3 (6.4)	1 (6.3)	1.000
Antihypertensives, n (%)			
ACEi	3 (6.4)	0 (0.0)	0.564
ARB	31 (66.0)	14 (87.5)	0.121
CCB	24 (51.1)	11 (68.8)	0.257
Beta-blocker	7 (14.9)	5 (31.3)	0.162
Diuretic	11 (23.4)	4 (25.0)	1.000
Other	2 (4.3)	0 (0.0)	1.000
Antidiabetic therapy, n (%)	10 (21.3)	3 (18.8)	1.000
Morning home SBP, mmHg	135.0 ± 14.0	133.2 ± 13.6	0.641
Morning home DBP, mmHg	83.7 ± 11.5	82.5 ± 14.0	0.766
Morning home HR, beats/min	68.8 ± 11.2	69.1 ± 9.1	0.916
CAVI, unit	9.0 ± 1.2	9.4 ± 1.8	0.409
baPWV, m/sec	19.5 ± 5.7	21.5 ± 8.3	0.369
ALT, U/L	18.7 ± 5.8	47.8 ± 18.9	<0.001
AST, U/L	23.4 ± 7.5	35.0 ± 14.0	0.005
Uric acid, mg/dL	7.7 ± 1.0	7.7 ± 1.7	0.952
Creatinine, mg/dL	0.91 ± 0.24	0.93 ± 0.18	0.768
eGFR, mL/min/1.73 m^2^	64.5 ± 16.9	65.7 ± 14.7	0.792
hs-CRP, ng/mL	802 (363, 1840)	1140 (424, 1985)	0.670
NT-pro BNP, pg/mL	66 (36, 143)	50 (21, 159)	0.554
UACR, mg/g·Cr	15.9 (8.2, 49.2)	15.0 (8.7, 46.9)	0.969
Cystatin-C, mg/L	1.05 ± 0.19	1.07 ± 0.30	0.753

Values are mean ± standard deviation, median (interquartile range), or number of patients (%). Abbreviations: ACEi, angiotensin-converting enzyme inhibitor; ARB, angiotensin II receptor blocker; CCB, calcium channel blocker; SBP, systolic blood pressure; DBP, diastolic blood pressure; HR, heart rate; CAVI, cardio-ankle vascular index; baPWV, brachial-ankle pulse wave velocity; ALT, alanine aminotransferase; AST, aspartate aminotransferase; eGFR, estimated glomerular filtration rate; hs-CRP, high sensitive C-reactive protein; NT-proBNP, amino terminal-pro B-type natriuretic peptide; UACR, urinary albumin-creatinine ratio.

**Table 3 biomedicines-11-00674-t003:** Baseline clinical characteristics of participants with alanine aminotransferase level <40 U/L or ≥40 U/L.

Variables	ALT <40 U/L (n = 55)	ALT ≥40 U/L (n = 8)	*p*-Value
Age, years	68.6 ± 7.2	65.3 ± 8.8	0.332
Male, n (%)	46 (83.6)	7 (87.5)	1.000
Body mass index, kg/m^2^	25.5 ± 3.3	28.0 ± 3.3	0.074
Smoking, n (%)	11 (20.0)	2 (25.0)	0.665
Drinking, n (%)	40 (72.7)	4 (50.0)	0.229
Medical history, n (%)			
Diabetes mellitus	11 (20.0)	2 (25.0)	0.665
Dyslipidemia	20 (36.4)	4 (50.0)	0.467
Chronic kidney disease	2 (3.6)	0 (0.0)	1.000
Liver disease	3 (5.5)	0 (0.0)	1.000
Stroke	1 (1.8)	1 (12.5)	0.240
Heart failure	2 (3.6)	0 (0.0)	1.000
Coronary artery disease	5 (9.1)	0 (0.0)	1.000
Non-valvular atrial disease	3 (5.5)	1 (12.5)	0.427
Antihypertensives, n (%)			
ACEi	3 (5.5)	0 (0.0)	1.000
ARB	38 (69.1)	7 (87.5)	0.421
CCB	28 (50.9)	7 (87.5)	0.066
Beta-blocker	11 (20.0)	1 (12.5)	1.000
Diuretic	13 (23.6)	2 (25.0)	1.000
Other	2 (3.6)	0 (0.0)	1.000
Antidiabetic therapy, n (%)	11 (20.0)	2 (25.0)	0.665
Morning home SBP, mmHg	133.9 ± 14.1	138.6 ± 11.6	0.325
Morning home DBP, mmHg	82.8 ± 11.3	87.0 ± 17.0	0.522
Morning home HR, beats/min	68.1 ± 10.6	74.4 ± 9.3	0.136
CAVI, unit	9.0 ± 1.2	9.7 ± 2.3	0.417
baPWV, m/sec	19.4 ± 5.4	24.2 ± 11.0	0.257
ALT, U/L	21.0 ± 7.9	61.1 ± 18.6	<0.001
AST, U/L	23.9 ± 7.3	43.3 ± 15.1	0.008
Uric acid, mg/dL	7.7 ± 1.1	7.9 ± 2.0	0.806
Creatinine, mg/dL	0.91 ± 0.23	0.93 ± 0.21	0.859
eGFR, mL/min/1.73 m^2^	64.7 ± 16.2	65.3 ± 17.8	0.942
hs-CRP, ng/mL	802 (375, 1930)	1140 (433, 1575)	0.942
NT-pro BNP, pg/mL	61 (33, 143)	62 (13, 159)	0.613
UACR, mg/g·Cr	14.7 (8.2, 42.4)	42.7 (13.6, 256.8)	0.148
Cystatin-C, mg/L	1.04 ± 0.19	1.18 ± 0.36	0.305

Values are mean ± standard deviation, median (interquartile range), or number of patients (%). Abbreviations: ACEi, angiotensin-converting enzyme inhibitor; ARB, angiotensin II receptor blocker; CCB, calcium channel blocker; SBP, systolic blood pressure; DBP, diastolic blood pressure; HR, heart rate; CAVI, cardio-ankle vascular index; baPWV, brachial-ankle pulse wave velocity; ALT, alanine aminotransferase; AST, aspartate aminotransferase; eGFR, estimated glomerular filtration rate; hs-CRP, high sensitive C-reactive protein; NT-proBNP, amino terminal-pro B-type natriuretic peptide; UACR, urinary albumin-creatinine ratio.

## Data Availability

The data presented in this study are available from the corresponding author upon reasonable request.

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
