# Peer review of "The Effects of Topiroxostat, a Selective Xanthine Oxidoreductase Inhibitor, on Arterial Stiffness in Hyperuricemic Patients with Liver Dysfunction: A Sub-Analysis of the BEYOND-UA Study"

_biomedicines, 2023, doi:10.3390/biomedicines11030674_

Round 1

Reviewer 1 Report

Fujishima et al. found that XO inhibitors improve CAVI in hyperuricemic individuals with liver dysfunction. It would be very interesting to limit the patient population of choice in hyperuricemic patients. However, some modifications may be necessary.

Major comments:

The authors separate the ASTs by 22, 33, and 40. How these values were derived might be discussed further. Also, the authors discuss NAFLD/NASH, but do not seem to provide details on the liver injury of individuals in this study. An addendum should be added as to whether there were other causes of liver damage.

Minor comments:

Hyperuricemia is defined as a serum uric acid level of more than 7.0 mg/dL, not serum uric acid level ≥ 7.0 mg/dL.

Author Response

Response to Reviewer #1

Reviewer comments for the author

Fujishima et al. found that XO inhibitors improve CAVI in hyperuricemic individuals with liver dysfunction. It would be very interesting to limit the patient population of choice in hyperuricemic patients. However, some modifications may be necessary.

Authors’ response

We thank you for your time and effort in reviewing our manuscript. We really appreciate beneficial suggestions to improve the quality of our manuscript. We have responded to your comments point by point as follows:

Major comments:

The authors separate the ASTs by 22, 33, and 40. How these values were derived might be discussed further. Also, the authors discuss NAFLD/NASH, but do not seem to provide details on the liver injury of individuals in this study. An addendum should be added as to whether there were other causes of liver damage.

Authors’ response

I greatly appreciate your valuable comments. The reason we chose the cut-off value of ALT 30 U/L, in addition to 22 U/L (the median value) and 40 U/L (the upper limit of normal), was that such a slight increase of ALT ≥30 U/L in Japanese subjects has been reported to be associated with lifestyle-related chronic liver diseases such as NAFLD (Tanaka K et al. Hepatol Res. 2014;44:1196-207.). In addition, as shown in Figure 1 for Reviewer, our previous study enrolling healthy volunteers and patients with type 2 diabetes (Kawachi Y et al. J Diabetes Investig. 2021;12:1512-1520.) revealed that subjects with ALT ≥30 U/L exhibited significantly higher levels of plasma XOR activity. Then, we have revised the manuscript as follows:

Page 11, Lines 316-320 (4. Discussion)

“In addition to the ALT cut off values of 22 U/L (the median value) and 40 U/L (the up-per limit of normal), we also analyzed our subjects by dividing them above or below 30 U/L because such a slight increase of ALT ≥30 U/L in Japanese subjects has been re-ported to be associated with lifestyle-related chronic liver diseases such as NAFLD [31].”

Regarding the complication of liver diseases, the BEYOND-UA study was not originally designed to focus on liver function. Thus, the underlying causes of liver injury in each subject were not examined in detail, and different types of liver disease, not only NAFLD, could be enrolled in this study. However, we have confirmed that there was one subject with chronic hepatitis C whose ALT level was 15 U/L and two subjects with alcoholic hepatitis whose ALT levels were 22 and 28 U/L in the topiroxostat group. We have addressed this important issue in the Discussion section as a study limitation and carefully revised the description of the term “NAFLD” throughout the manuscript as follows:

Page 2, Lines 56-64 (1. Introduction)

“Recently, we reported that circulating XOR in humans and mice markedly increased with elevations in liver enzymes such as serum alanine aminotransferase (ALT) or aspartate aminotransferase (AST), reflecting excessive leakage of hepatic XOR, and that topiroxostat, a selective XOR inhibitor, suppressed plasma XOR activity and attenuated the development of vascular neointima formation in a diet-induced mouse model of NAFLD/NASH [12,13]. Therefore, we hypothesize that XOR inhibitors may have the potential to prevent or delay cardiovascular complications, especially in patients with liver dysfunction, possibly beyond their UA-lowering effect.”

Page 10, Lines 307-310 (4. Discussion)

“Moreover, high XOR in liver disease conditions accelerated purine catabolism in the plasma per se using hypoxanthine secreted from vascular endothelial cells or adipocytes as substrate, which was accompanied by the development of vascular endothelial injury and neointimal proliferation [11-13].”

Page 12, Lines 383-388 (4. Discussion)

“Third, the BEYOND-UA study was not originally designed to focus on liver function; therefore, the underlying causes of liver injury in each subject were not examined in detail, and subjects with different types of liver disease, not only NAFLD, could be enrolled in this study. In fact, we have confirmed that there was one subject with chronic hepatitis C whose ALT level was 15 U/L and two subjects with alcoholic hepatitis whose ALT levels were 22 and 28 U/L in the topiroxostat group.”

Minor comments:

Hyperuricemia is defined as a serum uric acid level of more than 7.0 mg/dL, not serum uric acid level ≥ 7.0 mg/dL.

Authors’ response

We totally agree with your comments. It was a simple mistake. As pointed out by the reviewer, the authors have corrected the sentence as follows:

Page 1, Lines 42-44 (1. Introduction)

“Hyperuricemia, defined as a serum uric acid (UA) level of more than 7.0 mg/dL, is a potential risk factor for life-threatening complications, such as chronic kidney disease (CKD) and cardiovascular disease (CVD) [1-4].”

Reviewer 2 Report

In this manuscript the authors present the result of a subanalysis of the BEYOND-UA study to examine the effects of topiroxostat on arterial stiffness in hypertensive individuals with hyperuricemia. They report that topiroxostat use was associated to a decrease in cardioankle vascular index in the high ALT subgroup. The same subgroup showed high plasma XOR activity, that was suppressed by topiroxostat.The authors conclude that topiroxostat improved arterial stiffness parameters in hyperuricemic patients with liver dysfunction, an effect possibly related to its topiroxostat  inhibitory effect on plasma XOR. Overall, the paper has been analytically discussed. However the cohort studied and the further fractioning in small-numbered quartiles is difficult to understand, also in consideration that the statistically power of the study is not explained.

Major

Introduction line 2:  life-threatening vascular complications such as chronic kidney disease. Uric acid toxicity in the kidney is mainly tubulointerstitial, not vascular.

 The rationale of the cut offs values  for quartiles (and the need of using quartiles), in such a small number of patients,  is not clear. According to the Methods, quartiles included above or below 22 U/L (the median value), 30 U/L, or 40 U/L (the upper  limit of normal). From this sentence it appears that a lower limit quartile was not included. 

It is not clear how many subjects has abnormal AST/ALT. In table 1 the high ALT group was 35.8±17.6, indicating that some subject had ALT around 70 u.

Methods. It is unclear how many patients had liver dysfunction. It is unclear which kind of disease caused impared liver damage. Whas it NAFLD? How many subjects had Child Pugh A? This should be explained. In addition an increase in AST/ALT is more expressive of necrosis and liver mass than function; many CKD patients may have low AST/ALT and advanced-stage cirrhosis.

The cohort studied is small. The authors need to add information of the statistical power of the study. The ALT >30 Group was composed of only 16 subjects ( and ALT>40 only 8 subjects)-

In the discussion, the authors point to the NAFLD. Over the past decade, liver function tests including gamma-glutamyltransferase (GGT), alanine aminotransferase (ALT), alkaline phosphatase (ALP) and aspartate aminotransferase (AST) have emerged as markers of CVD risk in both population-based studies and patients with coronary artery disease(Atherosclerosis. 2009;202:11927, Atherosclerosis. 2014;236:7–17.PLoS One. 2014;9:e91410independent of their relationship with metabolic syndrome or non-alcoholic steatohepatitis (NASH). 

Author Response

Response to Reviewer #2

Reviewer comments for the author

In this manuscript the authors present the result of a subanalysis of the BEYOND-UA study to examine the effects of topiroxostat on arterial stiffness in hypertensive individuals with hyperuricemia. They report that topiroxostat use was associated to a decrease in cardioankle vascular index in the high ALT subgroup. The same subgroup showed high plasma XOR activity, that was suppressed by topiroxostat. The authors conclude that topiroxostat improved arterial stiffness parameters in hyperuricemic patients with liver dysfunction, an effect possibly related to its topiroxostat inhibitory effect on plasma XOR. Overall, the paper has been analytically discussed. However the cohort studied and the further fractioning in small-numbered quartiles is difficult to understand, also in consideration that the statistically power of the study is not explained.

Authors’ response

We thank you for your time and effort in reviewing our manuscript. We really appreciate beneficial suggestions to improve the quality of our manuscript. We have responded to your comments point by point as follows:

Major

#1 Introduction line 2: life-threatening vascular complications such as chronic kidney disease. Uric acid toxicity in the kidney is mainly tubulointerstitial, not vascular.

Authors’ response

We totally agree with your comments. As pointed out by the reviewer, the authors have removed the term “vascular” as follows:

Page 1, Lines 42-44 (1. Introduction)

“Hyperuricemia, defined as a serum uric acid (UA) level of more than 7.0 mg/dL, is a potential risk factor for life-threatening complications, such as chronic kidney disease (CKD) and cardiovascular disease (CVD) [1-4].”

#2 The rationale of the cut offs values for quartiles (and the need of using quartiles), in such a small number of patients, is not clear. According to the Methods, quartiles included above or below 22 U/L (the median value), 30 U/L, or 40 U/L (the upper limit of normal). From this sentence it appears that a lower limit quartile was not included.

Authors’ response

We apologize for any misunderstanding. As described in the Materials and Methods section, Lines 107-110, we analyzed our subjects by dividing them into two groups according to their baseline ALT levels, above or below 22 U/L (Table 1), above or below 30 U/L (Table 2), or above or below 40 U/L (Table 3), not using quartiles of baseline ALT. The reason we chose the cut-off value of ALT 30 U/L, in addition to 22 U/L (the median value) and 40 U/L (the upper limit of normal), was that such a slight increase of ALT ≥30 U/L in Japanese subjects has been reported to be associated with lifestyle-related chronic liver diseases such as NAFLD (Tanaka K et al. Hepatol Res. 2014;44:1196-207.). In addition, as shown in Figure 1 for Reviewer, our previous study enrolling healthy volunteers and patients with type 2 diabetes (Kawachi Y et al. J Diabetes Investig. 2021;12:1512-1520.) revealed that subjects with ALT ≥30 U/L exhibited significantly higher levels of plasma XOR activity. Then, we revised our manuscript as follows:

Page 3, Lines 107-110 (2. Materials and Methods)

“In the current sub-analysis, patients treated with topiroxostat were divided into two groups according to their baseline ALT level: above or below 22 U/L (the median value) (Table 1), above or below 30 U/L (Table 2), or above or below 40 U/L (the upper limit of normal) (Table 3) (Figure S1).”

Page 11, Lines 316-320 (4. Discussion)

“In addition to the ALT cut off values of 22 U/L (the median value) and 40 U/L (the up-per limit of normal), we also analyzed our subjects by dividing them above or below 30 U/L because such a slight increase of ALT ≥30 U/L in Japanese subjects has been re-ported to be associated with lifestyle-related chronic liver diseases such as NAFLD [31].”

#3 It is not clear how many subjects has abnormal AST/ALT. In table 1 the high ALT group was 35.8±17.6, indicating that some subject had ALT around 70 u.

Authors’ response

We appreciate the reviewer’s important comments. There were eight subjects with ALT levels above 40 U/L and six subjects with AST levels above 40 U/L, respectively. On the other hand, patients with severe liver dysfunction were not included in the present study due to an exclusion criterion for AST or ALT >2 times the upper limit of normal, leading to one of the major study limitations as described in the Discussion section (Page 12, Lines 388-391).

#4 Methods. It is unclear how many patients had liver dysfunction. It is unclear which kind of disease caused impared liver damage. Was it NAFLD? How many subjects had Child Pugh A? This should be explained. In addition an increase in AST/ALT is more expressive of necrosis and liver mass than function; many CKD patients may have low AST/ALT and advanced-stage cirrhosis.

Authors’ response

We totally agree with your comments. The BEYOND-UA study was not originally designed to focus on liver function. Thus, the underlying causes of liver injury in each subject were not examined in detail, and subjects with different types of liver disease, not only NAFLD, could be enrolled in this study. However, we have confirmed that there was one subject with chronic hepatitis C whose ALT level was 15 U/L and two subjects with alcoholic hepatitis whose ALT levels were 22 and 28 U/L in the topiroxostat group. As the reviewer pointed out, it also cannot be ruled out that there were subjects with low AST/ALT and advanced-stage cirrhosis. Therefore, we have addressed this important issue in the Discussion section as a study limitation and carefully revised the description of the term “NAFLD” throughout the manuscript as follows:

Page 2, Lines 56-64 (1. Introduction)

“Recently, we reported that circulating XOR in humans and mice markedly increased with elevations in liver enzymes such as serum alanine aminotransferase (ALT) or aspartate aminotransferase (AST), reflecting excessive leakage of hepatic XOR, and that topiroxostat, a selective XOR inhibitor, suppressed plasma XOR activity and attenuated the development of vascular neointima formation in a diet-induced mouse model of NAFLD/NASH [12,13]. Therefore, we hypothesize that XOR inhibitors may have the potential to prevent or delay cardiovascular complications, especially in patients with liver dysfunction, possibly beyond their UA-lowering effect.”

Page 10, Lines 307-310 (4. Discussion)

“Moreover, high XOR in liver disease conditions accelerated purine catabolism in the plasma per se using hypoxanthine secreted from vascular endothelial cells or adipocytes as substrate, which was accompanied by the development of vascular endothelial injury and neointimal proliferation [11-13].”

Page 12, Lines 383-388 (4. Discussion)

“Third, the BEYOND-UA study was not originally designed to focus on liver function; therefore, the underlying causes of liver injury in each subject were not examined in detail, and subjects with different types of liver disease, not only NAFLD, could be enrolled in this study. In fact, we have confirmed that there was one subject with chronic hepatitis C whose ALT level was 15 U/L and two subjects with alcoholic hepatitis whose ALT levels were 22 and 28 U/L in the topiroxostat group.”

#5 The cohort studied is small. The authors need to add information of the statistical power of the study. The ALT >30 Group was composed of only 16 subjects ( and ALT>40 only 8 subjects).

Authors’ response

We totally agree with your comments. Although the data used in the present study were generated in a randomized controlled clinical trial, the post hoc analysis was not pre-specified. The post hoc nature of this analysis means that potential sources of bias cannot be excluded. Therefore, the results obtained from this study need to be confirmed in a prospective clinical trial as described in the Conclusion section (Page 12, Lines 400-401). We have addressed this issue, including the small sample size, in the revised manuscript as follows:

Page 12, Lines 378-381 (4. Discussion)

“Although the data used in this study were generated in a randomized controlled clinical trial, the post hoc analysis was not pre-specified. Thus, potential sources of bias cannot be ruled out due to the post hoc nature of this analysis. Second, the results must be interpreted with caution due to the small sample size in each patient subgroup and due to the lack of a placebo control group.”

#6 In the discussion, the authors point to the NAFLD. Over the past decade, liver function tests including gamma-glutamyltransferase (GGT), alanine aminotransferase (ALT), alkaline phosphatase (ALP) and aspartate aminotransferase (AST) have emerged as markers of CVD risk in both population-based studies and patients with coronary artery disease (Atherosclerosis. 2009;202:11927, Atherosclerosis. 2014;236:7–17.PLoS One. 2014;9:e91410) independent of their relationship with metabolic syndrome or non-alcoholic steatohepatitis (NASH).

Authors’ response

We greatly appreciate the reviewer’s valuable comment. As plasma XOR activity increases significantly with elevations in liver enzymes such as ALT and AST, we also consider it a very important finding that elevations in these liver enzymes themselves, whether due to NAFLD or not, are potentially associated with CVD risk. Thus, we have cited these articles suggested by the reviewer and revised the manuscript as follows:

Page 10, Lines 302-304 (4. Discussion)

“Besides, certain studies have suggested that liver function tests themselves, including ALT, AST, gamma-glutamyltransferase (GGT), and alkaline phosphatase (ALP), can be potential CVD risk markers, independent of their relationship to NAFLD [27-29].”

Round 2

Reviewer 1 Report

The authors worked diligently and revised the manuscript according to the reviewers' comments. The manuscript was improved and concerns about publication were addressed.

Reviewer 2 Report

Thank you for your response.